# III-Nitride Short Period Superlattices for Deep UV Light Emitters

**Sergey A. Nikishin**

Nano Tech Center, Department Electrical and Computer Engineering, Texas Tech University, Lubbock, TX 79423, USA; sergey.a.nikishin@ttu.edu; Tel.: +1-806-834-8807



**Featured Application: Advanced infrared, visible, and ultraviolet light emitters.**

**Abstract:** III-Nitride short period superlattices (SPSLs), whose period does not exceed ~2 nm (~8 monolayers), have a few unique properties allowing engineering of light-emitting devices emitting in deep UV range of wavelengths with significant reduction of dislocation density in the active layer. Such SPSLs can be grown using both molecular beam epitaxy and metal organic chemical vapor deposition approaches. Of the two growth methods, the former is discussed in more detail in this review. The electrical and optical properties of such SPSLs, as well as the design and fabrication of deep UV light-emitting devices based on these materials, are described and discussed.

**Keywords:** III-nitrides; short period superlattices; light emitters

---

## 1. Introduction

The invention of semiconductor double heterostructure laser [1,2] and the concept of a semiconductor superlattice (SL) [3] can be considered as the foundation of modern semiconductor p–n junction-based light emitters, lasers, and light-emitting diodes. Double heterostructure (DHS) is a semiconductor "sandwich" where a layer of narrow band gap semiconductor (referred to as an active layer or a well, depending on its thickness) is placed between n-type and p-type layers (cladding layers) of the wide bandgap semiconductors. Under forward bias, the electrons and holes are injected from the cladding layers into an active layer, and well confined there. Confinement of these injected carriers leads to very effective radiative recombination in direct bandgap semiconductors.

The SL periodic layered structure of different crystalline semiconductors, allows for engineering of a bandgap of active (referred as a well) and cladding (referred as a barrier) layers when their thickness is of about a few nanometers. Note, these structures must be grown into two-dimensional (2D) growth mode, in order to get a flat/abrupt interface between all layers. The SLs based on the direct bandgap semiconductors in different combinations with p–n DHS are used for design and fabrication of light emitters operating in very wide range of wavelengths, ensuring the needs of optical communication, medicine, security, lighting, and agriculture [4–11]. We will discuss only p–n junction-based structures in this review.

III-Nitrides, AlN ($E_g$ = 6.2 eV, [12,13]), GaN ($E_g$ = 3.4 eV, [13]), InN ($E_g$ = 0.7 eV, [14–16]), and their AlGaInN, AlGaN, AlInN, InGaN alloys, are direct bandgap semiconductors which would significantly influence the development of new optoelectronic and light-emitting devices throughout the 21st century [17–19]. Most III-nitride (III-N) light-emitting and laser diodes contain different SLs with a period exceeding 4 nm [20–23]. These SLs, in addition to the bandgap engineering, allow for reducing the dislocation density propagating into an active layer of III-N light-emitting diodes (LEDs) grown on the heavily lattice-mismatched substrates, like silicon and sapphire [24–26]. Strain engineering in an active layer of LEDs allows for modification of an internal quantum efficiency of radiative carrier

recombination [27,28]. The design and fabrication of such light emitters, based on "long period" SLs, are well described in many books and reviews [29–36].

III-N LEDs based on very short period superlattices (SPSLs, sometimes referenced as digital alloys, DA), periodic structures of GaN/AlN, AlGaN/AlN, AlGaInN/AlN, InGaN/GaN, InN/GaN, and InAlN/GaN having a few monolayer thick wells and barriers, and a period not exceeding 2 nm, are very attractive for the design and fabrication of a new generation of light emitters. The main important difference between SPSLs and SL with a long period, is that carriers tunneling through barriers between quantum wells (QWs) in SPSLs already affect energy levels, and even lead to the formation of minibands (at least in the conduction band). Bandgap behavior of the InGaN/GaN, InN/GaN, and InAlN/GaN SPSLs, and their applications in visible and infrared light emitters, are well summarized in recent publications [37–40]. The GaN/AlN, AlGaN/AlN, and AlGa(In)N/AlN ($C_{In}$ < 0.02 mol fraction) SPSLs are very attractive for deep ultraviolet light emitters [41–48]. One of the attractive features of AlGa(In)N/AlN SPSLs relates to the formation of very sharp heterointerfaces over the entire range of compositions, which makes it possible to obtain well/barrier thicknesses comparable to the interatomic distance, and to make tunneling the main carrier transport mechanism. For InGaN/GaN structures, for example, the roughness of heterointerfaces increases if the composition of In approaches ~20%, which is important for practical applications, since carrier tunneling is difficult. This review aims to summarize the most significant efforts demonstrated in this field since 2002, when the first LED based on AlGa(In)N/AlN SPSLs operating at 280 nm, was demonstrated [41,42].

## 2. Growth and Structural Characterization

Most of III-N SPSLs for deep UV light emitters were grown using both molecular beam epitaxy (MBE) [41–43] and metal organic chemical vapor deposition (MOCVD) [37,44] methods on (0001) sapphire, Si (111), and (0001) GaN/sapphire template substrates. The detailed analysis of deep UV LED efficiency (internal and external), grown on different substrates, can be found in ref. [44].

One well-known advantage of MBE over MOCVD is the in situ monitoring of the growth process using the reflection high energy electron diffraction (RHEED) [49]. Analysis of the RHEED patterns in real time allows for controlling, at the monolayer (ML) scale, the structural properties of any substrate at the onset of epitaxy, the nucleation process and growth mode of the epitaxial layer, the growth rate of III-N compounds, and the composition of their alloys, by monitoring the period of RHEED intensity oscillations during deposition [50–52]. This statement can be illustrated by a few RHEED patterns.

The evolution of RHEED patterns illustrating the onset of gas source MBE (GSMBE) with ammonia on bulk (0001) AlN substrate is shown in Figure 1. As seen in Figure 1a, there are two types of reflections indicated separately by black solid and white dashed arrows. It was shown that this complex RHEED pattern can be attributable to the presence of $Al_2O_3$ surface islands [53]. The well-defined (00), (01), and (–01) reflections indicated by solid arrows can be attributed to the (1 × 1) AlN (0001) surface reconstruction at low temperatures. The weak additional reflections, indicated by dashed arrows, arise from formation of crystalline $Al_2O_3$ islands on the AlN surface. These islands cannot be removed by baking of the AlN substrate at high temperatures, up to ~1100 °C [53]. However, nitridation of such a surface, by exposing it to the flux of ammonia for a few minutes at a substrate temperature of ~800 °C, yields formation of a pure (1 × 1) surface structure on AlN (0001), as shown in Figure 1b. This (1 × 1) surface reconstruction was stable up to 900 °C. At this temperature, AlN, $Al_{0.6}Ga_{0.4}N$, and SPSL of AlN (3 ML)/$Al_{0.08}Ga_{0.92}N$ (3 ML), with a total of 100 pairs, were successfully grown. The entire SPSL was grown in the 2D mode, and formation of a (2 × 2) surface reconstruction is shown in Figure 1c. The surface was very flat, with the root mean square (rms) roughness of less than 1 nm, as measured by 1 × 1 μm$^2$ scans, using atomic force microscopy.

Figure 2 shows the evolution of the RHEED patterns illustrating 2D→3D→2D growth mode transitions during ammonia GSMBE of $Al_{0.55}Ga_{0.45}N$ (barrier)/$Al_{0.45}Ga_{0.55}N$ (well) structure on $Al_{0.55}Ga_{0.45}N$/AlN buffer grown on (0001) sapphire substrate [54,55].

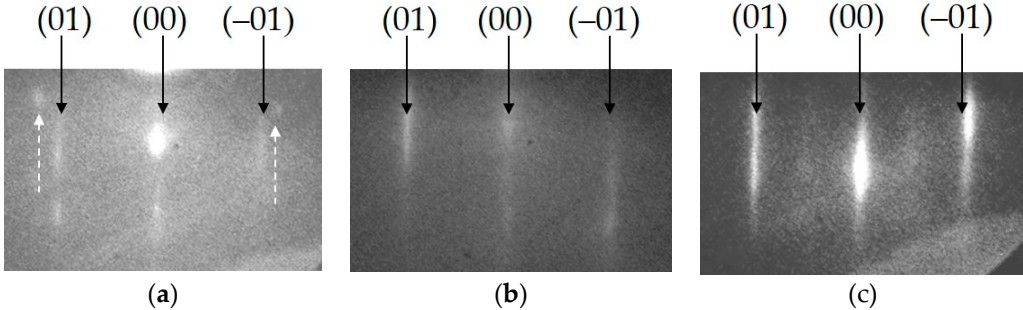

**Figure 1.** Evolution of reflection high energy electron diffraction (RHEED) patterns for different stages at onset of gas source molecular beam epitaxy (GSMBE). (**a**) The surface of (0001) AlN substrate at low temperatures. The (00), (01), and (–01) reflections from the (1 × 1) AlN surface and reflections from crystalline $Al_2O_3$ islands are indicated by arrows; (**b**) (1 × 1) surface reconstruction of AlN exposed to ammonia; (**c**) (2 × 2) surface structure after deposition of about 20 pairs of short period superlattices (SPSLs).

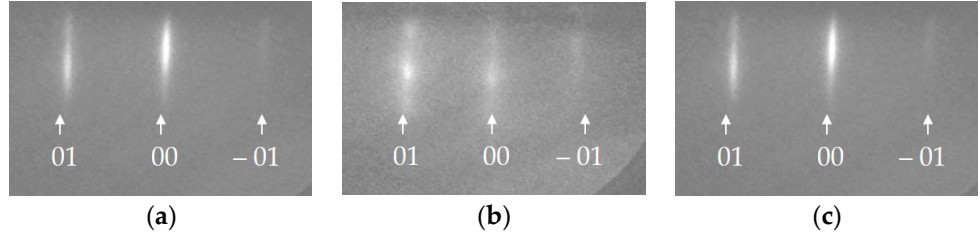

**Figure 2.** Evolution of RHEED patterns illustrating 2D and 3D growth modes at different (Al + Ga)/$NH_3$ flux ratios: (**a**) 2D-grown barrier at 20 sccm of ammonia; (**b**) 3D-grown well at 5.5 sccm of ammonia; (**c**) next 2D-grown barrier at 20 sccm of ammonia on a 3D-grown well.

All the barrier layers were grown in 2D growth mode with 1 × 1 surface reconstruction, as shown in Figure 2a, when an ammonia flux was sustained at 20 sccm. The wells grown under the same ammonia flux also demonstrated 2D growth mode. This mode was maintained at ammonia fluxes greater than 7 sccm (N-rich conditions), at a substrate temperature of 795 °C. With the ammonia flux reduced to 5.5 sccm, the RHEED patterns become quite spotty, as shown in Figure 2b. This behavior of the RHEED pattern is typical when the growth mode changes from 2D to 3D [56]. These growth conditions can be attributed to the "metal (Ga, Al)-rich" conditions, although additional experiments are required. It was shown that the barrier layer recovers when ammonia flux increased to 20 sccm and the RHEED pattern shows 2D growth mode, as shown in Figure 2c, by the time the next well is grown. Note that significant increase in the deep UV cathodoluminescence (CL) and photoluminescence (PL) emission from such grown structures was observed [57]. The increase was attributed to the formation of quantum dots (QDs) within the wells. It was concluded that the greatest CL intensity and longest PL lifetime for these structures are due to formation of quantum well (QW)/QD regions in $Al_xGa_{1-x}N/Al_yGa_{1-y}N$ (0.3 < x < 0.45, 0.53 < y ≤ 1) QW structures [57,58]. The approach described in [54,55,57,58] was recently adjusted for a plasma-assisted MBE (PAMBE), and successfully used in an active layer of deep UV LEDs emitting at 232 nm [59]. The emission at 219 nm from PAMBE-grown 2 ML thick GaN QDs was also observed [60].

Analyzing the state-of-the art results mentioned above and discussed in literature within last two to three years, we can conclude that one of the main current trends aimed at improving internal quantum efficiency (IQE) and external quantum efficiency (EQE) of UV LEDs is the creation of a low defective and highly efficient active layer in such structures. Future research should focus on finding the optimal ratio and distribution of QDs in the active layer, as well as on the development of the growth of LED structures on relatively inexpensive templates or bulk AlN substrates. Of course, it is much more effective to design and develop such an active layer using MBE, which provides in

situ monitoring of the growth process. Despite the fact that the MOCVD process dominates in the industrial growth of such structures, the results obtained using the MPE should make it possible to identify the most important structural and morphological factors influencing the radiative efficiency of the active layer of the LED. These new concepts can then be transferred to the MOCVD processes.

High resolution X-ray diffraction (HR-XRD) of AlN/AlGaN, AlN/GaN, and AlGaN/InGaN were carried out by many researchers [61–65] in order to estimate strain and dislocation density.

Note, HR-XRD measurements, in conjunction with Raman measurements, allow for estimation of the residual strain in SPSLs more precisely [66–69]. Usually, HR-XRD studies are carried out using a high-resolution diffractometer in double- and triple-axis alignment. A long range $2\theta$-$\omega$ scan of the (0002) reflection is shown in Figure 3 for a typical AlN/Al$_x$Ga$_{1-x}$N SPSL grown on (0001) Al$_{0.4}$Ga$_{0.6}$N/AlN/sapphire template. Individual peaks corresponding to the AlN and Al$_{0.4}$Ga$_{0.6}$N buffer layers, and the 0th, $\pm1$, and $\pm2$ satellites of the SPSL are well defined in Figure 3. The average SPSL composition, 0.68, was determined from the $2\theta$ position of the 0th peak [61]. From the position of the 0th and $\pm1$ satellite peaks, the average period of the SPSL was determined to be 2.236 nm. Using the experimentally determined period, and assuming well composition of Al$_{0.08}$Ga$_{0.92}$N and pure AlN barriers, the well and the barrier thicknesses are found to be 0.808 nm and 1.428 nm, respectively [61]. Simulations based on the experimentally determined SPSL parameters yield an excellent fit to the experimental data, as shown in Figure 3. The deviation of the well and barrier thicknesses from their integer lattice parameters (integer ML multiples) can be attributed to many factors, including residual strain in the SPSL, formation of interfacial layers, interface roughness, composition fluctuations in the well and barrier, stacking faults (SFs), and inversion domain boundaries (IDBs). The detailed analysis of significance of all these factors was conducted in reference [61].

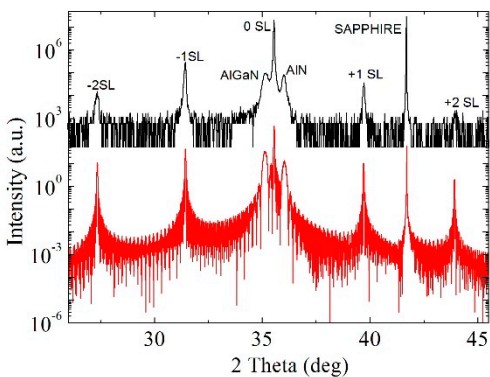

**Figure 3.** A long range $2\theta$-$\omega$ scan of (0002) reflection for AlN/Al$_x$Ga$_{1-x}$N ($0.07 < x < 0.09$) obtained using a hybrid X-ray mirror. Black line—data [61], red line—simulations (courtesy of Dr. A. Chandolu).

Crystalline microstructure of any semiconductor is a very important factor influencing the performance of all light emitters. Cross-sectional structure of SPSLs should be investigated by transmission electron microscopy (TEM), in order to get a nanoscale resolution. Two TEM cross-sections of AlN/AlGaN SPSL, grown by GSMBE, are shown in Figure 4. Although the growth conditions (substrate temperature, flux ratio, growth time) were the same, the crystalline quality of these SPSLs were very different. It is clearly seen that SPSL grown directly on sapphire substrate contains a very high density of inversion domain boundaries (IDBs). These domains start to grow from substrate/layer interface, mostly due to incomplete nitridation of sapphire at the onset of epitaxial growth [45,70]. It was shown [71] that IDBs dominate the light emission process in GaN containing these defects. A similar result was reported for MBE grown AlGaN/GaN SLs [72]. However, SPSLs with high density of IDBs have inferior electrical properties [45] and cannot be used in the preparation of light-emitting devices. The most detailed impact of sapphire nitridation on the formation of inverse domains in AlN layers grown by MOCVD was discussed in a recent paper [70].

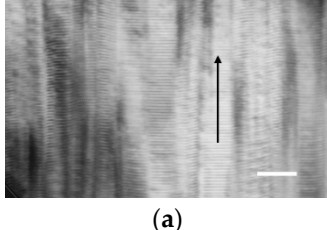 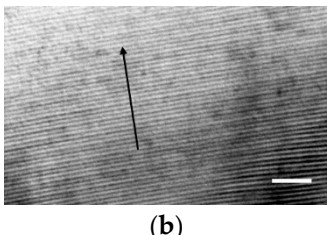

(**a**)  (**b**)

**Figure 4.** TEM cross-section of AlN/AlGaN SPSLs (courtesy of Dr. S. N. G. Chu) (**a**) grown directly on bare (0001) sapphire; (**b**) grown on ~50 nm thick AlN buffer layer. The white scale bars are 20 nm long. The [0001] direction is shown by black arrows.

## 3. Bandgap of AlN/AlGa(In)N SPSLs

The bandgap structure of III-Ns SPSLs convenient for the deep UV light emitters can be simulated using different software [73–75]. The BESST (Bandgap Engineering Superlattice Simulation Tool) commercially available package from STR Inc. [76] was used to analyze the results of different teams [77,78], as well as most of our experimental results. This software is suitable for modelling optoelectronic devices utilizing SPSLs as essential units of their heterostructure designs. The BESST calculates the SPSL electron and hole minibands using a tight-binding approach and numerical solution of the Schrödinger equation with account of complex valence band structure of III-nitride compounds. Coupled solution of the Poisson equation for electric potential, accounting for polarization charges at the heterostructure interfaces, and discrete drift-diffusion transport equations, allows building up the band diagram of a device at an arbitrary bias and calculating the corresponding electron and hole currents, as well as the radiative recombination rate and emission spectrum. Field-dependent mobilities of electrons and holes used in the transport equations are found, self-consistently, with the simulated minibands of SPSLs.

Fourier-transform infrared optical reflectance (FTIR) [79–82], photoluminescence (PL) [83–85], and cathodoluminescence (CL) [41,42,86] are widely used to estimate the effective bandgaps of AlN/Ga(Al,In)N SPSLs. At room temperature, these methods are mostly qualitative, although still very useful for express control and adjustment of light-emitter properties during fabrication. The room temperature FTIR and CL were successfully used to facilitate fabrication of the first deep UV LEDs operating at 280 nm using undoped and n- and p-type AlN/$Al_xGa_{1-x}$N SPSLs [41,42].

As an example, the experimental FTIR and CL effective bandgaps of AlN/$Al_{0.08}Ga_{0.92}$N SPSLs and simulations of these structures obtained using two different approaches [46,76] are shown in Figure 5. One can see that both simulations provide the slope of the optical energy gap dependence on the SPSL period (actually, on the AlN barrier "effective" width), similar to that obtained by CL, whereas FTIR data demonstrate a different slope. This may have originated from the fact that FTIR measures the spectral dependence of light absorption, which may involve higher electron minibands and lower hole minibands or levels, whereas luminescence occurs mainly from the ground state minibands, due to their dominant occupation. Simulations by BESST show that SPSLs, regarded here, possess two different electron minibands and up to three heavy- and light-hole minibands, which can be considered as single energy levels at large SPSL periods because of miniband narrowing. Hence, contribution of the extra minibands to the light absorption may explain the difference in the optical energy gap determination by FTIR and CL.

Of course, the Stokes shift, which is related to the excited state configuration in the well material, is a factor in all radiative recombination processes and, therefore, should also yield to red shift of the CL's estimated bandgap. Note that scatter in the experimental data shown in Figure 5 can be attributed to monolayer level uncertainty in the well and barrier thickness across the wafer, as well to local composition fluctuations in the well alloy [61,65].

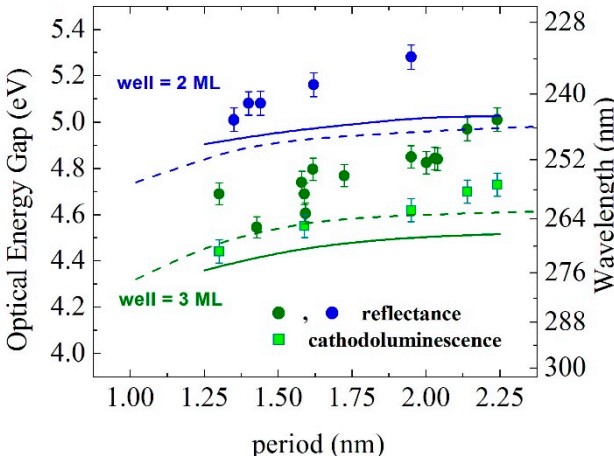

**Figure 5.** Experimental optical reflectance and cathodoluminescence (CL)-obtained effective bandgaps of AlN/Al$_{0.08}$Ga$_{0.98}$N SPSLs vs its period, grown with two nominal well thicknesses (2 and 3 monolayers (MLs), blue and green symbols, respectively) [45]. Theoretical simulations based on two approaches [46,76] are shown by continuous dashed and solid curves, respectively.

## 4. AlN/AlGa(In)N SPSL Doping and Ohmic Contacts

The efficiency of deep UV LEDs is very sensitive to the doping level of p- and n-type emitters. There are no significant issues with n-type doping of III-Ns compounds and their alloys, since Si, a common n-type dopant, behaves as a shallow donor, even in wide bandgap AlGaN alloys [87]. Although the activation energy of Si significantly increases in AlN [88–90], the resistivity of Si-doped AlN/Al$_x$Ga$_{1-x}$(In)N (0.05 < x < 0.1) SPSLs is very low, 0.015–0.040 $\Omega\cdot$cm, and the electron concentration exceeds $10^{19}$ cm$^{-3}$ [42,91]. Such n-type SPSLs can also be used as contact layers. Using a Ti/Al/Ti/Au stack annealed at 700 °C, specific contact resistance of the order of $10^{-5}$ $\Omega\cdot$cm$^2$ was obtained for AlN/AlGa(In)N SPSL with ~5.1 eV bandgap [92].

Unfortunately, for III-Ns compounds and their alloys, there is only one convenient p-type dopant, Mg, the experimentally determined activation energy of which varies from ~120 to ~220 meV in GaN [93,94], and reaches more than 500 meV in AlN [94,95]. A detailed analysis of the Mg activation energy in p-AlGaN epitaxial layers over the entire composition range was recently published [96]. It should be noted that if Mg-doped AlGa(In)N is grown using MOCVD, then Mg activation is required. This can be done by both rapid thermal annealing at elevated temperatures [97,98] and holding the sample under an electron beam irradiation [98–101]. It was also shown that Mg–O co-doping reduces acceptor activation energy in GaN [102,103], as in AlN [104]. However, this method of doping is not widely used since oxygen can react with aluminum and gallium, forming undesirable oxides of these metals, especially when used in molecular beam epitaxy.

The hole density at room temperature in Al$_x$Ga$_{1-x}$N:Mg alloys with Mg concentration at ~$10^{19}$–$10^{20}$ cm$^{-3}$ is shown in Figure 6a. Note that all Mg concentrations were obtained using secondary ion mass spectrometry (SIMS) [25,103].

A significant decrease in the hole concentration with an increase in the Al content in AlGaN is consistent with an increase in the activation energy of Mg in the wide bandgap layers. For AlN/Al$_x$Ga$_{1-x}$(In)N (0.03 < x < 0.08) SPSLs with 2–3 ML thick wells, and periods of the order of 6–8 MLs, the average AlN concentration in SPSLs can be changed in the range of y$_{ave}$ = 0.5–0.8. The average Mg concentration in these SPSLs is usually at the level of $10^{19}$ cm$^{-3}$. The concentration of holes can be at the level of $10^{18}$ cm$^{-3}$, even in SPSLs with high average AlN content, as is seen in Figure 6a. Such structures were obtained using both MBE and MOCVD methods [25,41,42,45,91,105–110].

Figure 6b shows the results of temperature-dependent Hall characterization of three SPSLs and one AlGaN layer. The Mg concentration was ~$10^{19}$ cm$^{-3}$ in SPSLs, and ~3 × $10^{19}$ cm$^{-3}$ in AlGaN.

Note that the composition of the well was in the range $0.03 < x < 0.08$ which is very similar to the Al concentration in the AlGaN thick layer. Average AlN concentrations of these SPSLs were $y_{ave}$ = 0.65, 0.55, and 0.45. The average SPSL compositions, $y_{ave}$, were obtained using X-ray diffraction [61]. The hole density, here, corresponds to the hole concentration averaged over the full SPSL thickness. Fitting lines 1, 2, and 3 were calculated using BESST simulator, assuming AlN barriers in the SPSL to be relaxed. It is clearly seen that the hole concentration varies very little with the temperature in SPSLs, and is highly dependent on the temperature (almost 1.5 orders of magnitude) in a uniformly Mg-doped $Al_{0.05}Ga_{0.95}N$ layer. Similar experimental results were reported by many different teams [45,105–115].

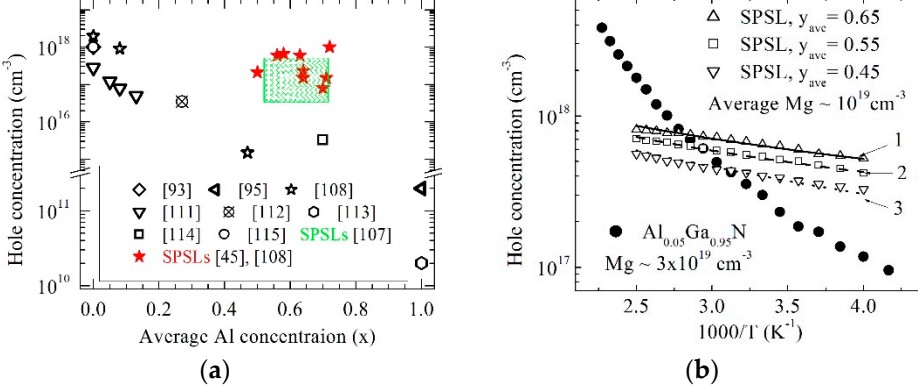

**Figure 6.** Hole concentration in $Al_xGa_{1-x}N$ alloys and AlN/AlGa(In)N SPSLs. (**a**) The dependence on average Al concentration in epitaxial layers (black symbols) and in SPSLs (red symbols and green box); (**b**) The temperature dependence of the hole concentration in $Al_{0.05}Ga_{0.95}N$ epitaxial layer (~0.5 μm thick, filled circles) and in AlN/$Al_xGa_{1-x}N$ SPSLs (open symbols). Lines 1, 2, and 3 are the results of the applied BESST simulator [76].

The valence band edge profile in AlN(~0.90 nm)/$Al_{0.03}Ga_{0.97}N$(~0.52 nm) SPSL, computed by the BESST simulator [76] at room temperature, are shown in Figure 7. The position of the Fermi level in this SPSL corresponds to the zero-energy horizontal line shown in Figure 7a. The acceptor levels in $Al_{0.03}Ga_{0.97}N$-well and AlN-barrier layers are indicated by the dashed lines.

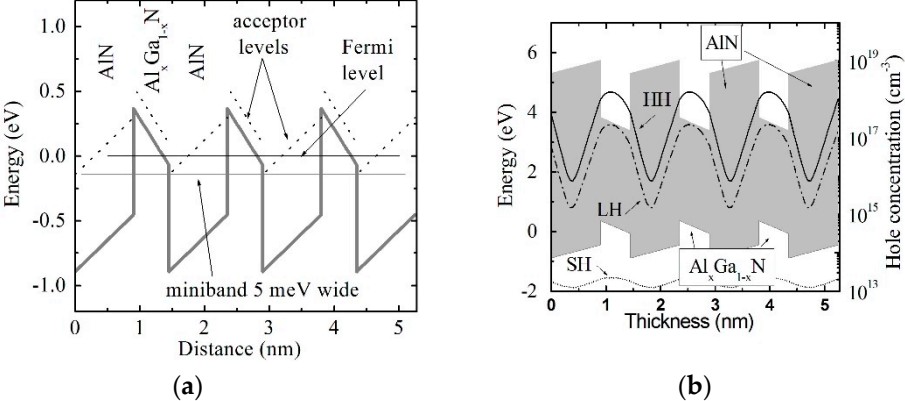

**Figure 7.** Valence band layout in AlN/$Al_{0.03}Ga_{0.97}N$ SPSL, computed for room temperature (**a**) and band diagram and heavy (HH), light (LH), and split-off (SH) hole concentrations in this sample (**b**). The acceptor level positions were estimated by neglecting the effect of the hole confinement in the quantum wells.

It is clearly seen that the acceptor levels in the quantum wells are above the Fermi level. Thus, these acceptors should be not activated. In the AlN barriers, the Fermi level crosses the acceptor energy levels, yielding a partial activation of acceptors in these layers. The degree of activation depends

on temperature, creating hole tunneling from barriers to wells, which results in the exponential dependences of hole concentration, as shown in Figure 6.

Figure 7b shows the distributions of the heavy, light, and split-off holes in the same SPSL obtained by self-consistent solution of the Poisson and Schrödinger equations [76], accounting for the complex valence band structure of III-nitrides [116]. It is clearly seen that the total hole density in such an SPSL mostly depends on the heavy hole concentration. The contribution of the split-off holes can be neglected, due to the fact that this subband in the SPSL is far below the heavy and light subbands. Note, the holes in the SPSL are accumulated in the wells, partially penetrating into the AlN barriers.

One could conclude that the efficiency of Mg activation is a very weak function of temperature in wide bandgap $AlN/Al_xGa_{1-x}(In)N$ SPSL ($E_g$ from ~4.2 to ~5.3 eV) [107,108]. Thus, these SPSLs are very attractive for fabrication of transparent low resistive ohmic contacts for deep UV light emitters.

Various contact metallization schemes have been applied for Mg-doped p-GaN, including Au, Ni, Ti, Pd, Pt, Au/Ni, Au/Pt, Au/Cr, Au/Pd, Au/Mg/Au, Au/Pt/Pd, Au/Cr/Ni, Au/Pt/Ni, Au/Ni/Pt, Au-Zn/Ni, Si/Ni/Mg/Ni, etc. [117]. Oxidized Au/Ni is the most common ohmic contact to p-GaN, due to the relatively low specific contact resistance ($\rho_c$) and simplicity of fabrication [118]. It was shown that formation of crystalline NiO and $Ni_3Ga_4$ play an essential role in the reduction of the specific contact resistance of p-GaN [118–121]. While the formation of NiO yields a reduction of the specific contact resistance, the formation of $Ni_3Ga_4$ results in the eventual increase in the contact resistance [119–121]. The lowest specific contact resistance $\rho_c$ ~$9.2 \times 10^{-6}$ and $2 \times 10^{-6}$ $\Omega$ cm$^2$ at 150 °C were achieved for p-type Mg-doped GaN and $Al_xGa_{1-x}N$ layers [121,122].

Similar approach was successfully used for fabrication of ohmic contacts on p-type $AlN/Al_xGa_{1-x}N$ SPSLs [108]. Figure 8 shows the temperature dependence of the specific contact resistances, $\rho_c$, for three SPSLs with average AlN content of $y_{ave}$ ~0.7 and for the ~300 nm thick Mg-doped $p-Al_{0.03}Ga_{0.97}N$ epitaxial layer. The Mg concentration in all samples was ~$10^{19}$ cm$^{-3}$ and room temperature $\rho_c$ was ~$4 \times 10^{-5}$ $\Omega \cdot$cm$^2$.

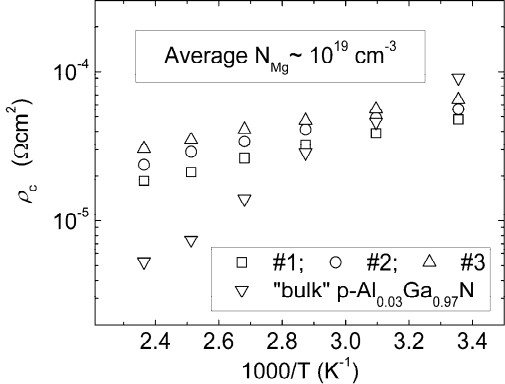

**Figure 8.** Specific contact resistances of Mg-doped $AlN/Al_xGa_{1-x}N$ SPSLs ($y_{ave}$ ~0.7) and $Al_{0.03}Ga_{0.97}N$ epitaxial layer at different temperatures.

It was shown that the temperature dependence of the specific contact resistance is primarily controlled by the activation energy of Mg acceptors in $Al_xGa_{1-x}N$ [123]. The $\rho_c$ of metal/SPSL ohmic contact, as seen in Figure 8, also follows the temperature dependence of hole concentration in these structures, which are very similar to that shown in Figure 6b, since the density of acceptors activated in AlN barriers is the primary factor influencing the hole concentration in the well, producing a weaker temperature dependence compared to bulk materials. Thus, a weak dependence of $\rho_c$ on temperature is expected, even when thermionic mechanism of current injection dominates in the metal/SPSL ohmic contact. This expectation agrees with the results shown in Figure 8.

## 5. Deep UV Light Emitters Based on AlN/AlGa(In)N SPSLs

The ability to change the bandgap in SPSL by several hundred meV, by changing the thickness of the well by 1 ML, opened up a very simple path to the implementation of the DHS LEDs. Since SPSL with particular barrier/well thicknesses has its effective bandgap, we can consider them to form a heterostructure in the sense that an SPSL-based active layer with a smaller effective bandgap is sandwiched between n- and p-type SPSL- based cladding layers with a larger bandgap.

A schematic cross-section of a typical DHS for a sub-300 nm LED is shown in Figure 9. The structure consists of six main parts: (1) thin, <100 nm, AlN buffer; (2) thick, ~200 nm, $Al_{0.7-0.8}Ga_{0.3-0.2}N$ buffer for reduction of dislocation density in a device structure (this layer can be replaced by AlN/GaN SPSL—"dislocation absorber"); (3) n-type AlN(5 ML)/$Al_{0.92-0.97}Ga_{0.08-0.03}$(In)N(2 ML) SPSL cladding layer of ~400 nm; (4) undoped AlN(5 ML)/$Al_{0.92-0.97}Ga_{0.08-0.03}$(In)N(3 ML) SPSL active region (5 pairs); (5) p-type AlN(5 ML)/$Al_{0.92-0.97}Ga_{0.08-0.03}$(In)N(2 ML) SPSL cladding layer of ~200 nm; (6) thin, ~2–5 nm, p-type contact layer with the composition of SPSL's well material or pure GaN:Mg. As can be seen from the TEM cross-section of this device, shown in the same Figure 9, the barrier/well interfaces are quite sharp, and the thickness of the well in the active region is greater than in the adjacent n- and p-type barriers.

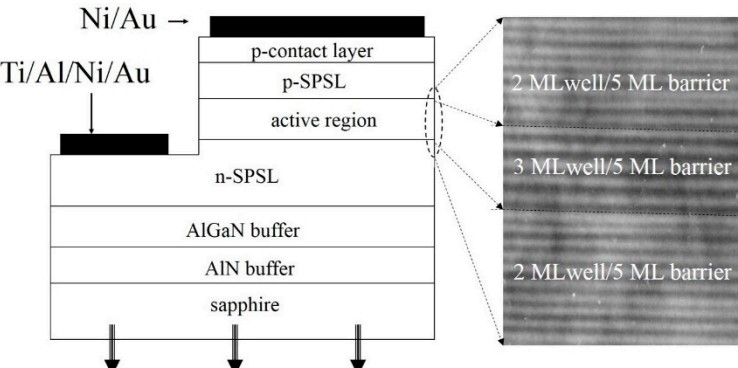

**Figure 9.** A not-in-scale schematic cross-section (on the **left**) of a double heterostructure (DHS) light-emitting diode (LED) and TEM cross-section (on the **right**) of the active region, and adjacent n- and p-type emitters of this diode.

The current–voltage (I–V) characteristic of a typical 260 nm DHS mesa-LED is shown in Figure 10a. The LED turns on at ~6.0 V, and has a relatively high differential series resistance $R_{ser}$ of $115 \pm 5\ \Omega$ under forward bias from 8 V to 12 V. The $R_{ser}$ of mesa diodes is the sum of the contact ($R_{cont}$), spreading ($R_{sp}$), and vertical ($R_{ver}$) resistances. The contact resistance of a 160 μm diameter diode, obtained from specific contact resistance, is $R_{cont}$ ~90 Ω. The estimated spreading resistance of this LED is $R_{sp}$ ~20 Ω, taking into account the resistivity of n-type AlGaN buffer layer (not shown in the inset of Figure 10a). The resistance of the etched part of the mesa, corresponding to transport across the SPSL layers, is $R_{ver} \approx \rho_1 \times (h/A)$ where $\rho_1$ is the perpendicular (along the growth direction) resistivity of the SPSL, $h$ is the height of the mesa (~300 nm, with the thickness of p-SL ~200 nm), and $A$ is the contact area. Finally, we obtain $R_{ver}$ ~5 Ω, resulting in $\rho_1$ ~50 Ω cm for a p-type SPSL. Comparing this to the in-plane conductivity ($\rho_2$) of the p-type SPSL obtained from Hall measurements, $\rho_2$ ~4 Ω cm, we obtain the conductivity anisotropy $\rho_1/\rho_2$ ~13. These simple considerations indicate relatively low $\rho_1$, considering the high AlN fraction in these SPSLs, and underscore the importance of reducing the contact resistance to p-type materials. However, it is also important to lower $R_{sp}$ by optimization of the buffer layer thickness and its resistivity. Similar I–V characteristics were reported by different teams for LEDs emitting near the same deep UV range of the wavelengths [42–46,91–128].

The typical electroluminescence spectra of DHS LEDs based on AlN/AlGa(In)N SPSLs are shown in Figure 10b. All these spectra were obtained on LEDs with a similar geometry, and with the same currents. A significant decrease in EQE is evident for devices emitting in ever deeper UV. The external

quantum efficiency (EQE) of such simple LEDs does not exceed 0.1%. The absence of the electron blocking layer is the main drawback [129] of the structure shown in Figure 9. In order to increase the efficiency of such LED structures, it was proposed to insert a few monolayers-thick AlN electron blocking layer between the active layer and the p-emitter [35,36].

Summarizing, it should be noted that several new approaches to the use of SPSLs in combination with QDs in the active layer of the LED have been proposed [60,130–132]. The most promising, from my point of view, is the idea set forth in the ref. [60], where, experimentally, deep UV emission at 219 nm from ultrathin MBE-grown GaN/AlN quantum structure was demonstrated. I believe in the success of this approach, since we observed a similar effect almost 10 years ago [54,55,57,58].

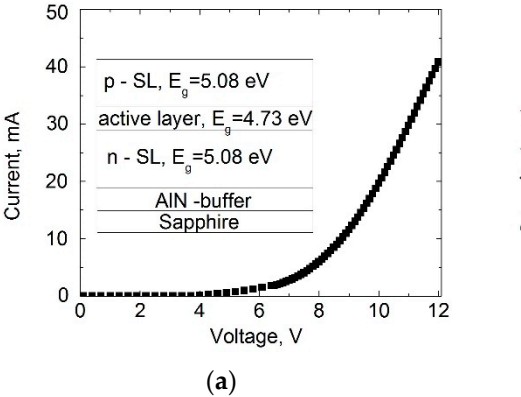 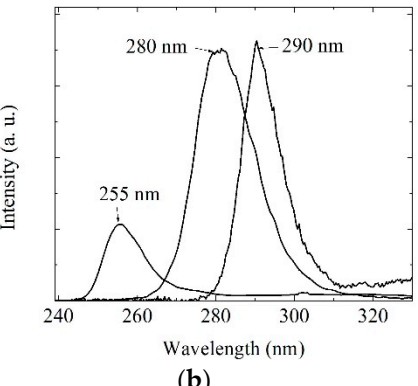

(**a**)　　　　　　　　　　　　　　　　　　　　(**b**)

**Figure 10.** Electrical and optical characteristics of AlN/AlGa(In)N SPSL-based DHS mesa-LEDs. (**a**) Current–voltage characteristic of a typical LED operating at 262 nm wavelength. Note that light emission was observed with forward *dc* current above 2 mA; (**b**) Electroluminescence spectra of similar AlN/AlGa(In)N SPSL-based DHS mesa-LEDs operating at 255, 280, and 290 nm.

## 6. Conclusions

It was demonstrated that AlN/Al$_x$Ga$_{1-x}$N p- and n-type SPSLs with average AlN content up to y$_{ave}$ ~0.7 and bandgap over 5.1 eV could have hole and electron concentrations exceeding $10^{18}$ cm$^{-3}$. Low-resistance ohmic contacts with specific contact resistance below ~$4 \times 10^{-5}$ Ω·cm$^2$ can be formed on these SPSLs. The LEDs based on SPSLs with emission wavelengths from 290 to 232 nm were demonstrated by different teams.

**Funding:** This paper received no external funding.

**Acknowledgments:** I would like to acknowledge all my colleagues involved in this research. I thank S. Yu. Karpov for very helpful discussions.

**Conflicts of Interest:** The author declares no conflict of interest.

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
