# Peer review of "III-Nitride Short Period Superlattices for Deep UV Light Emitters"

_applsci, doi:10.3390/app8122362_

Round 1
Reviewer 1 Report
The author is reviewing the topic of short period superlattices (SPSLs) made of nitride compounds and integrated in ultra violet light emitting devices (UV LEDs). He is focusing on works published after 2002, which is after the first publications on such UV LEDs. The review covers growth aspects, electrical and optical characterizations as well as simulations.
The topic is undoubtedly timely and of relevance since SPCLs are a key feature for the electrical injection in the devices and for reducing the crystalline defects related to the heteroepitaxy of the active layers. This review covers a significant part of the existing literature and provides methodological and experimental guidelines for whoever would like to fabricate such SPSLs structures. However, an explicit description on the progress made since 2002 and of the actual challenges in the field are missing. For this reason, I would recommend the publication of this review only after MAJOR revisions.
Main question/remark:
In the various sections of your review, could you identify some figure of merits for the SPSLs (e.g. dislocation densities, roughness, conductivity, internal quantum efficiency, etc) and comment on their evolution since 2002. So that the reader understand better the advances in the field and have an idea of where the research might go next.
Minor questions/remarks:
- line 35: The band gaps for InN and AlN are now known with a better accuracy than ±50 meV. Could you update these values?
- line 41: It is mentioned that SPSLs can reduce dislocation density. Could you provide order of magnitudes?
- line 65: It is mentioned that various substrates are used to grow the SPSLs. Could you be more specific? And maybe elaborate on which one provides the best results (although probably at a monetary cost).
- line 97: Could we say that the growth occurs in N rich conditions for an ammonia flux below 7 sccm and in metal rich for an ammonia flux above 7 sccm? Mentioning ammonia fluxes is a little involved for a reader not familiar with gas source molecular beam epitaxy.
- line 107: Are the quantum dots localization centers within the quantum wells? Is the case similar to the well reported case of InGaN QWs?
- line 107: At what temperature are done the PL measurements? Is the quantum dot still confining electron-hole pairs at room temperature?
- line 143: I would suggest the following recent reference for the nucleation of inversion domains: https://doi.org/10.1063/1.5008480.
- line 144: In GaN, the inversion domain boundary are localization centers at low temperature only, not at room temperature since they are not deep enough.
- line 145: Could you cite some work or provide elements for stating that inversion domain boundaries result in poorer electric properties?
- line 285: There are in this paragraph some redundancies with the introduction. For instance, the acronym DHS was already defined and a similar discussion on the advantages of molecular beam epitaxy was already done.
Author Response
I appreciate the reviewer for the detailed analysis of my manuscript. Please look over attached file.

Reviewer 2 Report
The author reviewed III-Nitride short period superlattice deep UV LEDs including material growth, doping, contacts, devices, etc. Overall the topic is well reviewed. However, the author missed several important aspects that should be included in the review paper. What is the state-of-the-art III-Nitride short period superlattice deep UV LED? Some major device parameters of this type of LEDs should be summarized, such as EQE, current-injection efficiency, operating voltages, etc. Device stability should be reviewed. What is the lifetime of these LEDs? What are major challenges? How does this type of deep UV LEDs perform compared to other deep UV LED technologies?
Author Response
I appreciate your comments. Please look over attached file containing my answers.

Round 2
Reviewer 1 Report
The author has nicely answered my questions.